# Structure and ion-release mechanism of $P_{IB-4}$-type ATPases

Christina Grønberg[1], Qiaoxia Hu[1], Dhani Ram Mahato[2], Elena Longhin[1], Nina Salustros[1], Annette Duelli[1], Pin Lyu[1,3], Viktoria Bågenholm[1], Jonas Eriksson[2], Komal Umashankar Rao[4], Domhnall Iain Henderson[4], Gabriele Meloni[5], Magnus Andersson[2], Tristan Croll[6], Gabriela Godaly[4], Kaituo Wang[1], Pontus Gourdon[1,7]*

[1]Department of Biomedical Sciences, University of Copenhagen, Copenhagen, Denmark; [2]Department of Chemistry, Umeå University, Umeå, Sweden; [3]Department of Sciences, University of Copenhagen, Copenhagen, Denmark; [4]Department of Laboratory Medicine, Lund University, Lund, Sweden; [5]Department of Chemistry and Biochemistry, The University of Texas, Dallas, United States; [6]Cambridge Institute for Medical Research, Department of Haematology, University of Cambridge, Cambridge, United Kingdom; [7]Department of Experimental Medical Science, Lund University, Lund, Sweden

**Abstract** Transition metals, such as zinc, are essential micronutrients in all organisms, but also highly toxic in excessive amounts. Heavy-metal transporting P-type ($P_{IB}$) ATPases are crucial for homeostasis, conferring cellular detoxification and redistribution through transport of these ions across cellular membranes. No structural information is available for the $P_{IB-4}$-ATPases, the subclass with the broadest cargo scope, and hence even their topology remains elusive. Here, we present structures and complementary functional analyses of an archetypal $P_{IB-4}$-ATPase, sCoaT from *Sulfitobacter* sp. NAS14-1. The data disclose the architecture, devoid of classical so-called heavy-metal-binding domains (HMBDs), and provide fundamentally new insights into the mechanism and diversity of heavy-metal transporters. We reveal several novel P-type ATPase features, including a dual role in heavy-metal release and as an internal counter ion of an invariant histidine. We also establish that the turnover of $P_{IB}$-ATPases is potassium independent, contrasting to many other P-type ATPases. Combined with new inhibitory compounds, our results open up for efforts in for example drug discovery, since $P_{IB-4}$-ATPases function as virulence factors in many pathogens.

*For correspondence: pontus@sund.ku.dk

Competing interest: The authors declare that no competing interests exist.

## Editor's evaluation

This paper presents crystal structures of sCoaT, a heavy metal transporting P-type ATPase. These structures and complementary functional data define the overall fold of this protein and provide insight into several mechanistic features, including a conserved histidine proposed to act as a novel counter-ion during transport. This work will be of interest to biochemists and microbiologists interested in the transport of transition metals, structural biology of membrane proteins and drug development.

## Introduction

The ability to adapt to environmental changes in heavy-metal levels is paramount for all cells, as these elements are essential for a range of cellular processes and yet toxic at elevated concentrations (*Waldron et al., 2009*; *Kozlowski et al., 2009*). Transition metal transporting P-type ($P_{IB}$) ATPase

**eLife digest** Heavy metals such as zinc and cobalt are toxic at high levels, yet most organisms need tiny amounts for their cells to work properly. As a result, proteins studded through the cell membrane act as gatekeepers to finetune import and export. These proteins are central to health and disease; their defect can lead to fatal illnesses in humans, and they also help bacteria infect other organisms.

Despite their importance, little is known about some of these metal-export proteins. This is particularly the case for $P_{IB-4}$-ATPases, a subclass found in plants and bacteria and which includes, for example, a metal transporter required for bacteria to cause tuberculosis. Intricate knowledge of the three-dimensional structure of these proteins would help to understand how they select metals, shuttle the compounds in and out of cells, and are controlled by other cellular processes.

To reveal this three-dimensional organisation, Grønberg et al. used X-ray diffraction, where high-energy radiation is passed through crystals of protein to reveal the positions of atoms. They focused on a type of PIB-4-ATPases found in bacteria as an example.

The work showed that the protein does not contain the metal-binding regions seen in other classes of metal exporters; however, it sports unique features that are crucial for metal transport such as an adapted pathway for the transport of zinc and cobalt across the membrane. In addition, Grønberg et al. tested thousands of compounds to see if they could block the activity of the protein, identifying two that could kill bacteria.

This better understanding of how PIB-4-ATPases work could help to engineer plants capable of removing heavy metals from contaminated soils, as well as uncover new compounds to be used as antibiotics.

proteins are critical for cellular heavy-metal homeostasis, providing efflux of for example copper, zinc, and cobalt from the intracellular milieu. Indeed, malfunctioning of the human $P_{IB}$-members, ATP7A and ATP7B, cause the fatal neurological Menkes disease and Wilson disease (***Bull et al., 1993***; ***Vulpe et al., 1993***). The $P_{IB}$-ATPases belong to the P-type ATPase superfamily of integral membrane proteins, which exploit energy from ATP hydrolysis for transport of cargo across cellular membranes. These proteins share an overall mechanism described by the so-called Post-Albers cycle (***Albers et al., 1963***; ***Post and Sen, 1965***), as established by decades of structural and functional investigations of primarily $Ca^{2+}$-, $Na^+/K^+$-, and $H^+$-specific P-type ATPases (***Toyoshima et al., 2000***; ***Toyoshima and Nomura, 2002***; ***Toyoshima et al., 2004***; ***Olesen et al., 2004***; ***Olesen et al., 2007***; ***Winther et al., 2013***; ***Toyoshima et al., 2013***; ***Morth et al., 2007***; ***Shinoda et al., 2009***). In summary, four cornerstone states, E1–E1P–E2P–E2, provide alternating access and affinity for the transported ions (and counterions, if present). Inward facing (e.g. cytosolic) E1 and outward facing (e.g. extracellular) E2P conformations are coupled to ATP-dependent phosphorylation (yielding ion-occluded E1P) and dephosphorylation (to occluded E2) of an invariant catalytical aspartate, respectively.

$P_{IB}$-ATPases are subdivided into groups based on conserved sequence motifs and the selectivity towards transported transition metal ions (***Smith et al., 2014***; ***Argüello, 2003***; ***Zielazinski et al., 2012***; ***Zhitnitsky and Lewinson, 2014***). Whereas $Cu^+$- and $Zn^{2+}$-transporting $P_{IB-1}$ and $P_{IB-2}$ ATPases are relatively well characterized, little is known regarding the $P_{IB-4}$ proteins, which comprise some of the simplest and shortest proteins within the entire P-type ATPase superfamily (***Smith et al., 2014***). They are present in plants, archaea, and prokaryotes, and have been assigned a role as virulence factors in pathogens, as for example the $P_{IB-4}$-ATPase MtCtpD is required for tuberculosis infections (***Sassetti and Rubin, 2003***; ***Joshi et al., 2006***), and therefore represent attractive targets for novel antibiotics.

The $P_{IB-4}$-ATPases are classically referred to as cobalt transporters. However, the metal specificity of the $P_{IB-4}$-ATPases remains elusive as some members have a confirmed cobalt specificity, while others seemingly have broader or altered ion transport profiles, also transporting ions such as $Zn^{2+}$, $Ni^{2+}$, $Cu^+$, and even $Ca^{2+}$ (***Zielazinski et al., 2012***; ***Raimunda et al., 2012***; ***Moreno et al., 2008***; ***Scherer and Nies, 2009***; ***Seigneurin-Berny et al., 2006***). Thus, the $P_{IB-4}$-ATPases appear to have the widest scope of transported ions of the $P_{IB}$-ATPases, and it is possible that further subclassification principles and sequence motifs will be identified. Due to the broad ion transport range, they have been proposed to

serve as multifunctional emergency pumps that can be exploited under extreme environmental stress to maintain heavy-metal homeostasis (*Smith et al., 2017*).

Hitherto, the available high-resolution structural information of full-length $P_{IB}$-ATPases is limited to two structures each of ion-free conformations of the $Cu^+$-transporting $P_{IB-1}$-ATPase from *Legionella pneumophila* (LpCopA) (*Gourdon et al., 2011b*; *Wang et al., 2014*), and the $Zn^{2+}$-transporting $P_{IB-2}$-ATPase from *Shigella sonnei* (SsZntA) (*Wang et al., 2014*). Thus, the principal architecture of the $P_{IB-4}$-ATPases remains debated, as sequence analyses have proposed different topologies for the N-terminus: with or without (1) the so-called HMBDs, and (2) the first two transmembrane helices, MA and MB (*Smith et al., 2014*; *Andersson et al., 2014*; *Rosenzweig and Argüello, 2012*; *Drees et al., 2015*), which both are present in other $P_{IB}$-ATPases (*Figure 1—figure supplement 1a*). These represent structural features that have been suggested to be important for ion-uptake and/or regulation in other $P_{IB}$-ATPases (*Gourdon et al., 2011b*; *Wang et al., 2014*; *González-Guerrero and Argüello, 2008*; *Mattle et al., 2013*), raising questions if similar levels of protein control are absent or replaced in the $P_{IB-4}$ group. In addition, despite a shared overall architecture, the $P_{IB-1}$ and $P_{IB-2}$ structures suggested significantly different types of entry and exit pathways, hinting at unique translocation mechanisms for each $P_{IB}$ group (*Sitsel et al., 2015*). However, it remains unknown if similar molecular adaptions have taken place in $P_{IB-4}$-ATPases to handle the unique array of cargos. To address these fundamental questions, we determined structures of a $P_{IB-4}$-ATPase in different states and validated our findings using in vitro functional characterization.

## Results and discussion
### Metal specificity

We employed the established $P_{IB-4}$ model sCoaT (UniProt ID A3T2G5) to shed further light on the structure and mechanism of the entire $P_{IB-4}$-class. As the metal ion specificity of the $P_{IB-4}$-ATPases is known to be wide, the ATPase activity was assessed in vitro in lipid–detergent solution using the so-called Baginski assay, in the presence of a range of different heavy metals. The protein exhibited clear $Zn^{2+}$- and $Cd^{2+}$-dependent ATPase activity, while $Co^{2+}$ only stimulated ATP hydrolysis at high ion concentrations (*Figure 2—figure supplement 1*). This is in partial agreement with the ion range profile previously reported for sCoaT, as higher $Co^{2+}$ sensitivity has been detected using different functional assays and different experimental conditions (*Zielazinski et al., 2012*; *Figure 2—figure supplement 1*).

The fact that the $K_M$ value for the $Co^{2+}$-dependent sCoaT activity reported previously is lower than measured in this study is unexpected (*Figure 2—figure supplement 1b*; *Zielazinski et al., 2012*). We therefore assessed if this observation relates to lower available concentration of $Co^{2+}$ consequent to chelation by buffer solution components, or if this metal interferes with the colour development in the ATPase assays determining $P_i$ concentrations (*Figure 2—figure supplement 1c*). However, $Co^{2+}$ and $Zn^{2+}$ display similar Baginski colour development as determined by calibration with separate standard curves. Moreover, neither exclusion of azide and molybdate to avoid possible $Co^{2+}$ binding of these compounds, nor supplementation of the reducing agent TCEP (to avoid possible oxidation of $Co^{2+}$ from molecular oxygen) has a significant effect on turnover. We also investigated if the type of assay may affect the outcome (*Figure 2—figure supplement 1c*). However, employment of the alternative Malachite Green Phosphate Assay essentially reproduced the relative activity in the presence of $Zn^{2+}$ and $Co^{2+}$, respectively (*Lanzetta et al., 1979*). sCoaT is purified in a buffer containing 5 mM β-mercaptoethanol, and even following dilution into the assay buffer the concentration is still approximately 100 μM, and as thiols can act as ligands for $Co^{2+}$ it may explain part of the differences in the $K_M$ values. However, this still does not explain why $Zn^{2+}$- and $Cd^{2+}$-dependent ATPase activity has not been observed for sCoaT in the previously study (*Smith et al., 2017*), although other $P_{IB-4}$-members have been associated with $Zn^{2+}$ activity. While not detected, the reported $K_M$ and $V_{max}$ may nevertheless be influenced by numerous environmental factors not tested for here, such as lipids, detergents, presence/absence/location of metal-binding his-tags, or other settings.

Despite that higher sensitivity has been measured for $Zn^{2+}$ compared to $Co^{2+}$, it cannot be excluded that $Co^{2+}$, rather than $Zn^{2+}$, is the preferred cargo in vivo as the relative intracellular availability of $Co^{2+}$ is more than three orders of magnitude higher than that of $Zn^{2+}$ in certain bacterial cells (*Osman et al., 2019*).

## Structure determination

We determined structures of sCoaT in metal-free conditions supplemented with two different phosphate analogues, $BeF_3^-$ and $AlF_4^-$, respectively, which previously have been exploited to stabilize E2 reaction intermediates of the transport cycle of $P_{IB}$-ATPases (*Wang et al., 2014*; *Gourdon et al., 2011b*; *Andersson et al., 2014*). The structures were determined at 3.1 and 3.2 Å resolution, using molecular replacement (MR) as phasing method and SsZntA as search model, and the final models yielded $R/R_{free}$ of 24.4/26.8 and 21.8/25.5 (*Table 1*). The two crystal forms were obtained using the HiLiDe method (crystallization in the presence of high concentrations of detergent and lipids) (*Gourdon et al., 2011a*). Surprisingly however, the crystal packing for both structures reveal only minor contacts between adjacent membrane-spanning regions, which are critical for the crystals obtained of most other P-type ATPase proteins (*Wang et al., 2014*; *Gourdon et al., 2011b*; *Sørensen et al., 2006*; *Figure 1—figure supplement 2*). Hence, some crystal forming interactions likely take place through lipid–detergent molecules. To our knowledge, this is the first time that type I crystals with unrestrained transmembrane domains are reported, but a consequence is that peripheral parts of the membrane domain are less well resolved (*Figure 1—figure supplement 3*). While this caused difficulty in modelling some transmembrane (TM) helices, satisfying solutions were found with the aid of the software ISOLDE (*Croll, 2018*) due to its use of AMBER forcefield which helped to maintain physical sensibility in the lowest resolution regions. In addition, root means square deviation, secondary structure as well as centre-of-mass of the transmembrane helices, only showed minor variation over time in MD simulations, indicative of a stable structure (*Figure 1—figure supplements 4 and 5*). The TM helices also showed lowered backbone root mean square fluctuation compared to more dynamic regions, such as the soluble domains and loop regions (*Figure 1—figure supplement 4b*).

## Overall structure, without classical HMBD

Examination of the structures reveals that the $P_{IB-4}$-ATPase architecture is reminiscent to that of other P-type ATPases, with three cytosolic domains, A (actuator), N (nucleotide-binding), and P (phosphorylation), as well as a membrane-spanning M-domain (*Figure 1a*). Furthermore, the core of

**Table 1.** Data collection and refinement statistics.

Statistics for the highest resolution shell are shown in parentheses.

| | E2-BeF$_3^-$ | E2-AlF$_4^-$ |
|---|---|---|
| **Data collection** | | |
| Wavelength (Å) | 1.0 | 1.0 |
| Space group | P 21 21 2 | P 21 21 2 |
| Cell dimensions | | |
| $a, b, c$ (Å) | 89.0 94.5 128.8 | 89.6 93.7 128.3 |
| $a, b, g$ (°) | 90 90 90 | 90 90 90 |
| Resolution (Å) | 47.3–3.1 (3.22–3.11) | 45.6–3.3 (3.37–3.25) |
| $R_{merge}$ (%) | 11.4 (276.3) | 15.5 (246) |
| $I / \sigma I$ | 17.8 (1.12) | 8.5 (0.98) |
| $CC_{1/2}$ | 1 (0.475) | 0.99 (0.37) |
| Completeness (%) | 97.3 (99.8) | 99.2 (99.9) |
| Redundancy | 13.3 (13.8) | 6.1 (6.6) |
| | | |
| **Refinement** | | |
| Resolution (Å) | 47.3–3.1 (3.22–3.11) | 45.6–3.3 (3.37–3.25) |
| No. reflections | 19,643 (1963) | 17,466 (1714) |
| $R_{work} / R_{free}$ (%) | 24.4/26.8 | 21.8/25.5 |
| *No. of atoms* | | |
| Protein | 4,695 | 4,695 |
| Ligand/ion | 5 | 6 |
| Water | 10 | 0 |
| *Average B-factors* | | |
| Protein | 135.91 | 152.54 |
| Ligand/ion | 84.15 | 86.47 |
| Solvent | 79.62 | |
| *R.m.s. deviations* | | |
| Bond lengths (Å) | 0.004 | 0.003 |
| Bond angles (°) | 0.77 | 0.83 |
| *Ramachandran statistics* | | |
| Favoured (%) | 97.8 | 96.9 |
| Allowed (%) | 2.2 | 3.1 |
| Outliers (%) | 0.0 | 0.0 |
| Clashscore | 1.05 | 7.89 |
| MolProbity score | 0.85 | 1.62 |

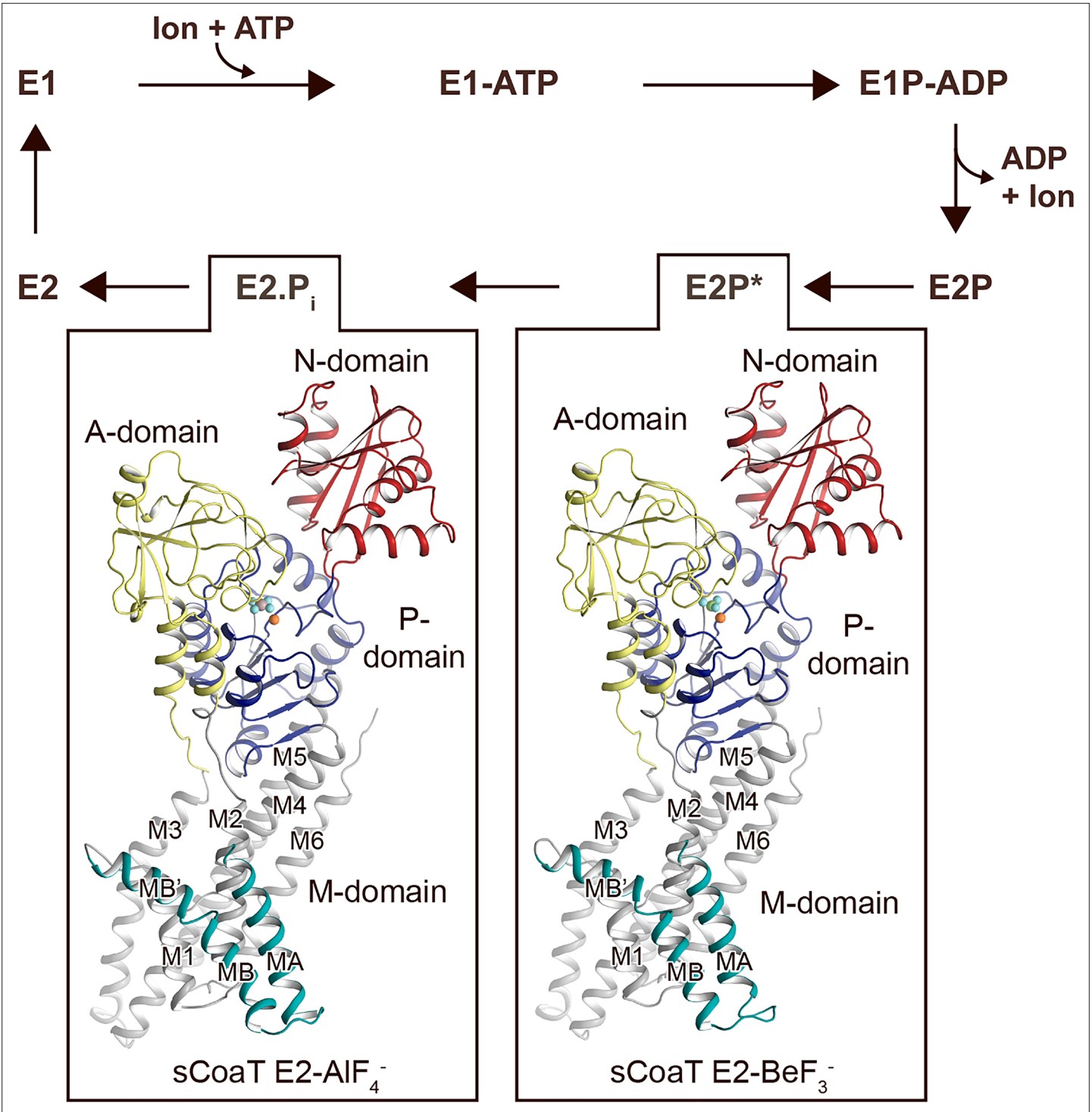

**Figure 1.** Overall architecture and reaction cycle. The sCoaT structures reveal that $P_{IB-4}$-ATPases comprise soluble A-, P-, and N-domains, shown in yellow, blue, and red, respectively, as well as a transmembrane domain with eight helices: MA and MB, in cyan, and M1–M6, in grey, and that the $P_{IB-4}$-topology lacks classical so-called heavy-metal-binding domain. The transport mechanism of P-type ATPases depends on ATP-dependent phosphorylation and auto-dephosphorylation, and includes four principal conformations, E1, E1P, E2P, and E2, where P denotes phosphorylation. The determined structures are trapped in two transition states following ion release – an occluded late E2P (E2P*) and an occluded transition state of dephosphorylation, $E2.P_i$.

The online version of this article includes the following figure supplement(s) for figure 1:

**Figure supplement 1.** Topology comparison.

*Figure 1 continued on next page*

*Figure 1 continued*

**Figure supplement 2.** Crystal packing of sCoaT E2-AlF$_4^-$ compared to the E2-BeF$_3^-$ crystal form of ZntA from *Shigella sonnei* (SsZntA, PDB ID: 4UMV).

**Figure supplement 3.** Electron density quality.

**Figure supplement 4.** Stability of the M-domain.

**Figure supplement 5.** Secondary structure stability of the M-domain.

the soluble portions, including the nucleotide-binding pocket and catalytic phosphorylation site at D369, are well conserved.

The topology of P$_{IB-4}$-ATPases has been a conundrum as sequence analyses have proposed different arrangements, with variable number of transmembrane segments and different sizes of the N-termini (*Smith et al., 2014*; *Gourdon et al., 2011b*; *Wang et al., 2014*; *Rosenzweig and Argüello, 2012*; *Drees et al., 2015*; *Andersson et al., 2014*). However, our data unambiguously demonstrate that P$_{IB-4}$-ATPases possess eight transmembrane helices, MA and MB followed by M1–M6. As previously observed for P$_{IB-1}$- and P$_{IB-2}$-ATPases, MB is kinked by a conserved Gly–Gly motif (G82 and G83), forming an amphipathic 'platform', MB', immediately prior to M1, see further below (*Figure 1—figure supplement 3*).

Are then HMBDs present in P$_{IB-4}$-ATPases as in the other P$_{IB}$ subclasses? As only the first 47 residues remain unmodelled in the final structures (*Table 1*), it is clear that many P$_{IB-4}$-ATPases including sCoaT are lacking a classical HMBD ferredoxin-like fold (typically 70 residues long). In agreement with this observation, the cysteine pair (CGIC in the sequence) in the N-terminus of sCoaT is rather positioned in MA, facing M1 (*Figure 1—figure supplements 1 and 3*), in contrast to the surface-exposed, metal-binding CXXC hallmark-motif detected in classical HMBDs. Functional analysis of mutant forms lacking these cysteines in vitro also support that they are unimportant for function (*Figure 2a*). We note that there are P$_{IB-4}$-ATPases with extended N-termini that, in contrast to sCoaT, may harbour HMBDs (*Smith et al., 2014*). Conversely, the sCoaT N-terminus is rich in metal-binding methionine, cysteine, histidine, aspartate, and glutamate residues, and this feature is conserved among P$_{IB-4}$-ATPases (*Figure 2—figure supplement 2*). We therefore explored the role of this N-terminal tail through assessment of an sCoaT form lacking the first 33 residues. However, in vitro characterization suggests only minor differences compared to wild-type, indicating that the residues upstream of MA are not essential for catalytic activity (*Figure 2a*). Aggregated, this hints at that no classical HMBD is present, and hence that this level of regulation is absent in many P$_{IB-4}$-ATPases, although it cannot be excluded that the N-termini are important in vivo.

Interestingly, it has been shown that the in vivo transport specificity of the sCoaT homolog from *Synechocystis PCC 6803* (CoaT) can be switched from Co$^{2+}$ to Zn$^{2+}$ by exchanging the N-terminal region to that of the Zn$^{2+}$ transporting P$_{IB-2}$ ATPase ZiaA from same organism (*Borrelly et al., 2004*). This demonstrates that P$_{IB-4}$-ATPases not only in vitro (our data), but also in vivo are able to transport Zn$^{2+}$, if the M-domain gain access to the metal. One possible explanation for the change of specificity for the CoaT chimeric construct is that the N-terminal peptide tail, as also suggested for ATP7B (*Yu et al., 2017*), prevents ATP hydrolysis through binding to the soluble domains, and this inhibition is then released upon binding of the cognate metal to the N-terminal and/or HMBD. However, it is also possible that the role of the N-terminal region of P$_{IB-4}$ proteins is to impair Zn$^{2+}$ acquisition, an ability that is lost when exchanged with the N-terminal part of ZiaA. Preliminary assessment of the metal specificity influence of the N-terminal tail of sCoaT suggests it has little or no effect on distinguishing between Co$^{2+}$ and Zn$^{2+}$ in vitro (*Figure 2—figure supplement 1d*). From this it is clear that further studies are needed to shed light on the function of the N-terminal region in P$_{IB}$-ATPases, also in P$_{IB-4}$-ATPases.

Associated, this raises questions also on the role of the above-mentioned MB' platform, which has been proposed to serve as an interaction site for HMBDs in P$_{IB-1}$- and P$_{IB-2}$-ATPases, and for the Cu$^+$-ATPases as a docking site for metal delivering chaperones (*Gourdon et al., 2011b*; *Wang et al., 2014*; *González-Guerrero and Argüello, 2008*; *Morin et al., 2009*). As there are no known zinc/cadmium chaperones for P$_{IB-4}$-ATPases, and because classical HMBDs appear to be missing in at least some proteins of the group, the MB' function may need to be revisited. Alternatively, the N-terminus may have merely been maintained through evolution without conferring functional benefits or disadvantages.

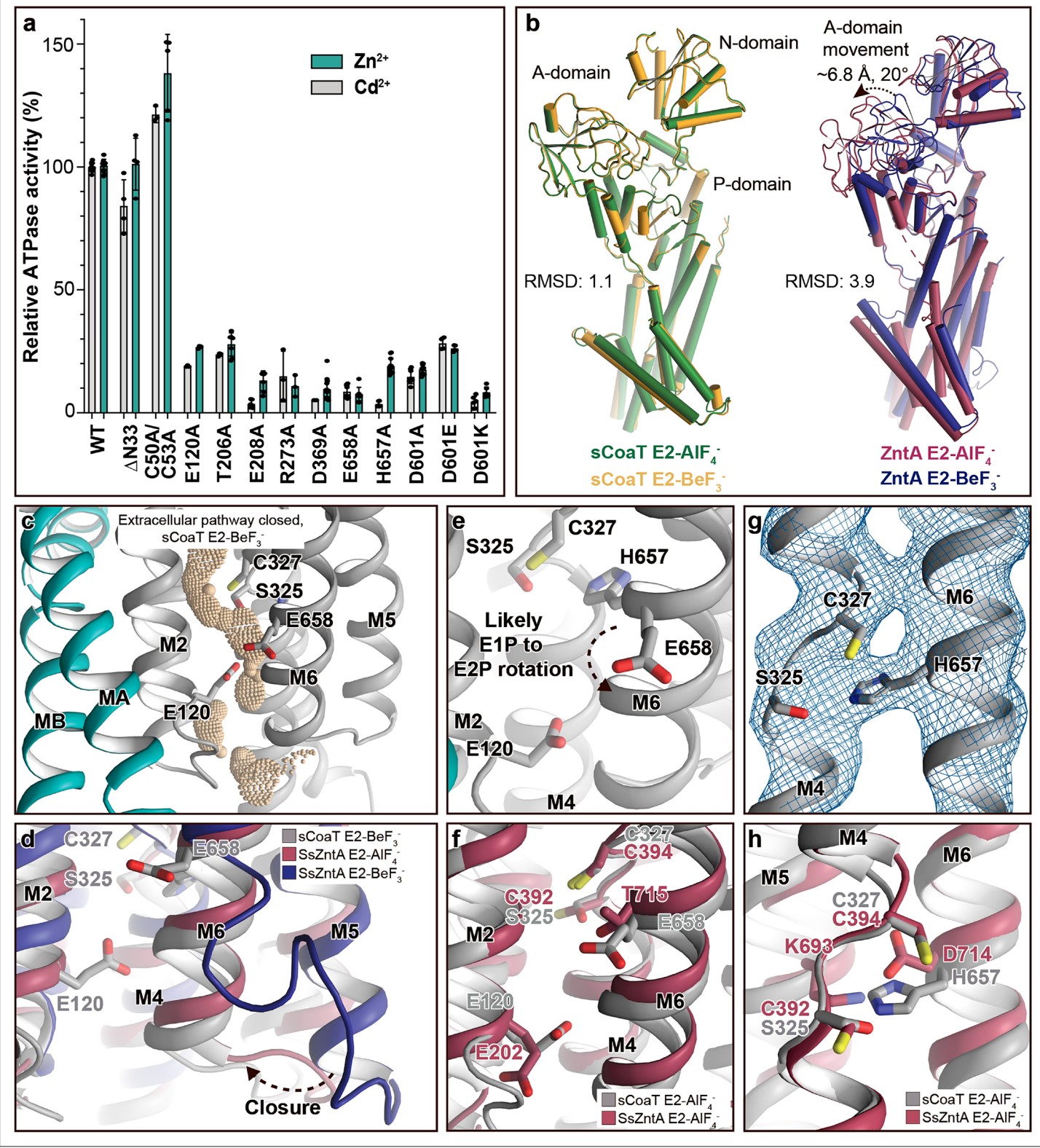

**Figure 2.** Mechanistic insight into the function of P$_{IB-4}$-ATPases. (**a**) Functional ATPase assay in lipid–detergent solution with targeted residues in sequential order. The wild-type (WT)-specific activity using the employed experimental conditions in the presence of 50 µM metal is 1.00 ± 0.01 µmol mg$^{-1}$ min$^{-1}$ with Zn$^{2+}$ and 2.80 ± 0.06 µmol mg$^{-1}$ min$^{-1}$ with Cd$^{2+}$, comparable to the activity previously measured for P$_{IB-4}$-ATPases. For biological averages and SD, see *Figure 2—figure supplement 1e*. (**b**) Comparisons of E2-AlF$_4^-$ and E2-BeF$_3^-$ structures of sCoaT and the equivalent of SsZntA (PDB ID of SsZntA structures: 4UMV and 4UMW). All superimpositions were performed based on the P-domain, and the RMSD values for the overall

*Figure 2 continued on next page*

*Figure 2 continued*

structures are indicated. (**c**) Identified cavity (wheat) in the E2-BeF$_3^-$ structure using the software HOLE. The E2-BeF$_3^-$ and the E2-AlF$_4^-$ (not shown) structures are occluded, lacking continuous connection between the ion-binding site to the outward environment. (**d**) The conformational changes that likely allow for closure of the release pathway, as illustrated from the E2-BeF$_3^-$ structure of SsZntA to the E2-AlF$_4^-$ structures of sCoaT or SsZntA. (**e–h**) Close views of ion-binding and -release residues in the M-domain of sCoaT and SsZntA. (**e**) The orientation of E658 is incompatible with high-affinity binding, and is likely contributing to ion release. (**f**) Release likely takes place via E658 and E120. (**g**) The sandwiched position between S325 and C327 of H657, including the final 2Fo-Fc electron density (blue). (**h**) The position of H657 in sCoaT overlaps with the one of K693 in SsZntA, and both likely serve as in-built counterions.

The online version of this article includes the following figure supplement(s) for figure 2:

**Figure supplement 1.** Metal selectivity screening and reproducibility.

**Figure supplement 2.** Sequence alignment of selected P$_{IB}$-ATPases.

**Figure supplement 3.** Comparison of E2 states overall and close views of the phosphorylation site.

**Figure supplement 4.** A-domain differences.

## Structures in a transition state of dephosphorylation

The classical view of P-type ATPases is that the E2P state is outward open and that the following transition state of dephosphorylation, E2.P$_i$, is occluded, and that these conformations can be stabilized using the phosphate analogues employed here for structure determination, BeF$_3^-$ and AlF$_4^-$, respectively. Furthermore, distinct ion-release pathways have been proposed among P$_{IB}$-ATPases (*Gourdon et al., 2011b*; *Wang et al., 2014*; *Andersson et al., 2014*; *Mattle et al., 2015*), including a narrow exit pathway lined by MA, M2, and M6 that remains open also in the E2.P$_i$ state for the P$_{IB-1}$-ATPases. In contrast, a wide opening extending from the location of the bound metal in the M-domain of ion-occluded states to the non-cytoplasmic side has been observed for the P$_{IB-2}$-ATPases, and this group becomes reoccluded with the E2P to E2.P$_i$ shift.

Surprisingly however, analysis of the two obtained structures suggests that the anticipated significant domain reorientations are absent in sCoaT (*Figure 2b*), and the models are in contrast rather similar. The compact assembly of the soluble domains and the position of the A-domain near the P-domain, placing the conserved TGE motif responsible for dephosphorylation towards the phosphorylation site, are typically associated with commencement of dephosphorylation, indicating that the two structures are trapped in an E2.P$_i$ like transition state (*Figure 2—figure supplement 3a, c, e*). This observation differs from the equivalent structures of the other structurally determined P$_{IB}$-ATPases, in which the phosphorylation site of the E2P state (stabilized by BeF$_3^-$) is shielded from the TGE loop as also observed for the well-studied sarcoendoplasmic reticulum Ca$^{2+}$-ATPase (SERCA) (*Figure 2—figure supplement 3d*).

Notably, analogous highly similar BeF$_3^-$- and AlF$_4^-$-stablized structures have recently also been observed for the Ca$^{2+}$-specific P-type ATPase from *Listeria monocytogenes* (LMCA1) (*Hansen, 2020*). It was proposed that LMCA1 preorganizes for dephosphorylation already in a late E2P state (E2P*, stabilized by BeF$_3^-$), in accordance with its rapid dephosphorylation. Favoured occlusion and activation of dephosphorylation directly upon ion release may thus also be the case for sCoaT, and consequently the E2-BeF$_3^-$ structure captured here may represents a late (or quasi) E2P state (E2P*).

Comparisons of the sCoaT structures to the equivalent structure of SsZntA (E2.P$_i$) revealed a unique arrangement of the A-domain (*Figure 2—figure supplement 4*). The TGE-loop region superposes well with the corresponding area in SsZntA, but the rest of the A-domain is rotated towards the P-domain – approximately 14° and 5.3 Å (*Figure 2—figure supplement 4*). However, it cannot be excluded that this rotation is due to crystal contacts as the two peripheral β-sheets of the A-domain are interacting tightly with parts of a neighbouring molecule. Additionally, we noticed that the A-domain of sCoaT possesses a surface-exposed extension similar to SERCA, but this feature is not present in P$_{IB-1}$- and P$_{IB-2}$-ATPases and it is not a conserved property in the P$_{IB-4}$ group either (*Figure 2—figure supplements 2 and 4*). Conversely, the M-domains of the two sCoaT structures are overall similar and appear outward occluded (*Figure 2c*), as also supported by comparisons with the equivalent structures of SsZntA, again contrasting to the situation observed in P$_{IB-1}$- and P$_{IB-2}$-ATPases.

## Ion release

Next, to shed light on ion release, we compared the sCoaT structures to the E2P state of SsZntA, in which the extracellular ends of M5 and M6 shifts away from the proposed metal-binding site, allowing an exit pathway to be formed (*Figure 2d*). Considering that $P_{IB-2}$- and $P_{IB-4}$-ATPases have overlapping cargo range, share overall topology and that they release ions in free form to the extra-cellular environment, in contrast to their $P_{IB-1}$ counterparts, we find it likely that they employ similar exit pathways, lined primarily by M2, M4, M5, and M6 (*Figure 2d*; *Wang et al., 2014*; *Andersson et al., 2014*).

The high-affinity-binding site in $P_{IB-4}$-ATPases has previously been suggested to be formed by residues from the conserved SPC- (starting from S325) and HEGxT- (from H657) motifs of M4 and M6, based on X-ray absorption spectroscopy and mutagenesis studies (*Zielazinski et al., 2012*; *Patel et al., 2016*). An outstanding remaining question is, however, how the ion is then discharged to the extracellular site? Among the resides that likely constitute the high-affinity-binding site, remarkably E658 of M6 is pointing away from the ion-binding region around the SPC motif (*Figure 2e* and *Figure 1—figure supplement 3*). We anticipate that E658 rotates away from its ion-binding configuration in the E1P to E2P transition, thereby assisting to lower the cargo affinity to permit release via the M2, M4, M5, and M6 cavity (*Figure 2e*). The conserved E120 of M2 (sometimes replaced with an aspartate in $P_{IB-4}$-ATPases) is located along this exit pathway. The residue also overlays with the conserved E202 in SsZntA (*Figure 2f*), which has been suggested to serve as a transient metal ligand, stimulating substrate release from the CPC motif of $P_{IB-2}$-ATPases (*Wang et al., 2014*). We propose a similar role for E120 in sCoaT as further supported by the decreased activity of E120A sCoaT form (*Figure 2a*).

## A unique internal counterion principle

Many P-type ATPases couple ion- and counter transport, and hence the reaction cycle cannot be completed without counterions. The importance of the counter transport has been demonstrated in for example $Ca^{2+}/H^+$- (such as SERCA), $Na^+/K^+$-, and $H^+/K^+$-ATPases (*Moller et al., 2010*; *Faxén et al., 2011*; *Abe et al., 2018*; *Dyla et al., 2020*). In contrast, the absence of counter transport has been proposed for $P_{IB-2}$-ATPases (*Wang et al., 2014*), $H^+$-ATPases (*Pedersen et al., 2007*), and P4-ATPases (*Nakanishi et al., 2020*), which rather exploit a built-in counterion. Specifically for the $P_{IB-2}$-ATPases, a conserved lysine of M5 (K693 in SsZntA) serves as the counterion, through interaction with the conserved metal-binding aspartate of M6 (D714 in SsZntA) in E2 states. Similarly, $P_{IB-1}$-ATPases are not $Cu^+/H^+$ antiporters, but a likely built-in counterion residue is not conserved in the group (*Abeyrathna et al., 2020*). Instead, it is possible that the requirement for counterion translocation is prevented by the narrow exit pathway, preventing backtransfer of the released ion and perhaps rendering complete occlusion unnecessary (*Abeyrathna et al., 2020*). For the $P_{IB-4}$-ATPases, biochemical studies have proposed an ion-binding stoichiometry of one (*Zielazinski et al., 2012*; *Raimunda et al., 2012*; *Smith et al., 2017*; *Patel et al., 2016*), however no information is available regarding the presence or absence of counter transport.

In the E2-BeF$_3^-$ sCoaT structure, we identify a tight configuration of HEGxT-motif H657, being sandwiched between the SPC residues, distinct from the M5 lysine–M6 aspartate interaction observed in $P_{IB-2}$-ATPases (*Figure 2g, h*). Despite the packing issues of the generated crystals, clear electron density is visible for H657, indicating a rigid conformation (*Figure 2g*). Moreover, activity measurements of an alanine substitution of H657 demonstrate that it is crucial for function (*Figure 2a*). In light of these findings and an earlier report suggesting that a mutation of the equivalent of H657 in MtCtpD leaves the ion affinity unaffected (*Patel et al., 2016*), we suggest this histidine serves as an internal counterion, similarly as for the invariant lysine in SsZntA, perhaps preventing backtransfer of released ions and for charge stabilization, however we cannot exclude that H657 is also part of the high-affinity-binding site in sCoaT.

The rigid conformation observed for H657 in the E2-BeF$_3^-$ structure is also observed in the E2-AlF$_4^-$ structure (*Figure 1—figure supplement 3b*). In contrast, for SsZntA the interaction between K693 and D714 is only detected in the E2.P$_i$ state. Thus, the interaction pattern is consistent with the idea that sCoaT preorganizes for dephosphorylation already in the (late) E2P state, with the associated occlusion and internal counter–ion interaction taking place earlier than for SsZntA.

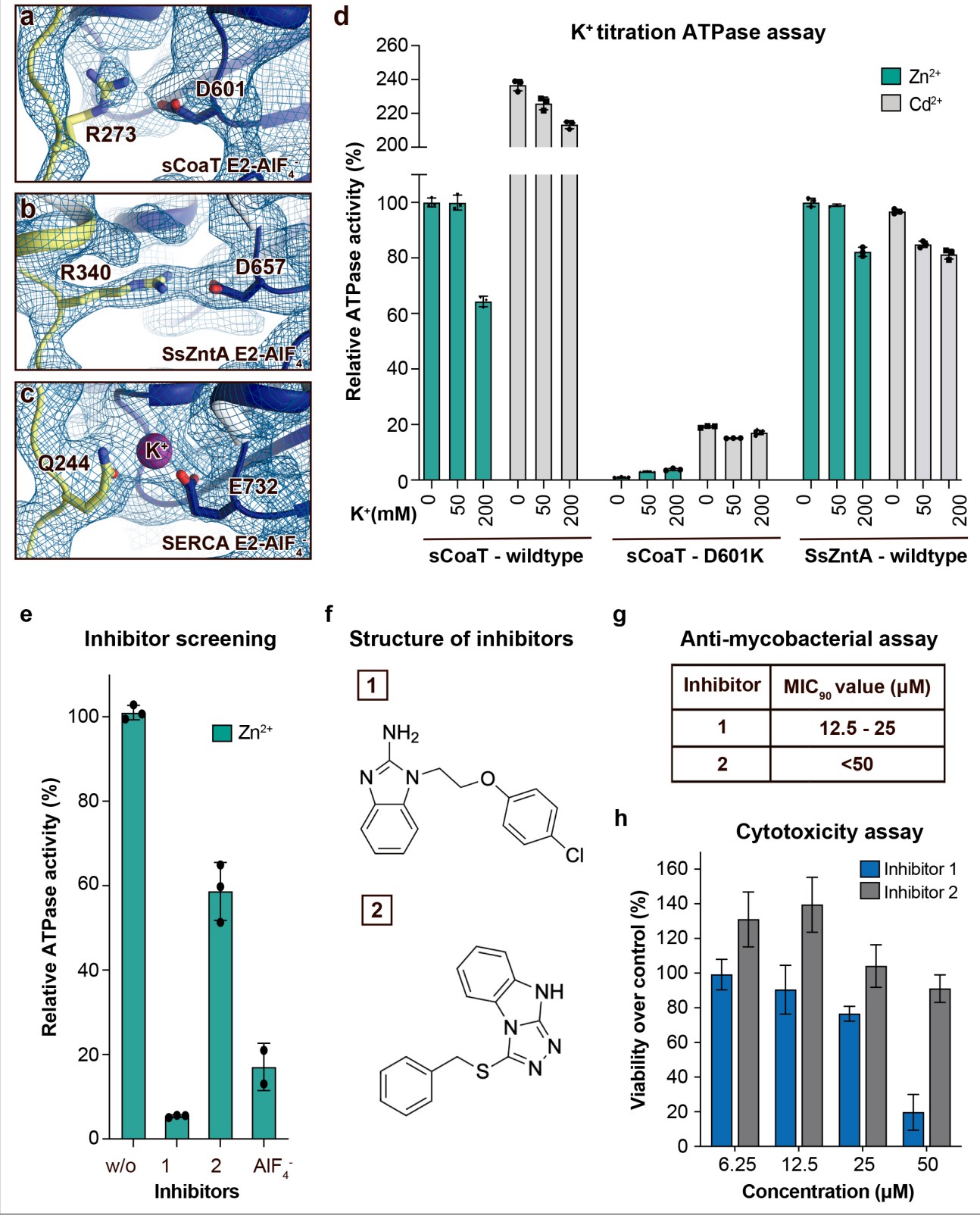

**Figure 3.** Regulation and inhibition. (**a–c**) Close views of the regulatory point-of-interaction between the A- and P-domains in the E2-AlF$_4^-$ structures of sCoaT, SsZntA, and SERCA (PDB IDs 4UMW and 1XP5) with the corresponding 2Fo-Fc electron density shown at $\sigma$ = 1.0 (blue mesh). (**a**) sCoaT (coloured as in *Figure 1*) with interaction between D601 and R273. (**b**) SsZntA (shown as panel a) with interaction between D657 and R340. (**c**) SERCA (shown as in panel a) with bound K$^+$ (purple) between E732 and Q244. (**d**) Functional ATPase assay in lipid–detergent solution of sCoaT (wild-type and D601K

*Figure 3 continued on next page*

*Figure 3 continued*

forms) as well as SsZntA (wild-type), using protein samples purified in the absence of $K^+$ and $Na^+$ (see Methods). The mean + SD of technical replicates is shown ($n$ = 3). KCl leaves the function of sCoat and SsZntA essentially unaffected in the presence of $Zn^{2+}$ (cyan) or $Cd^{2+}$ (grey). The equivalent form of sCoaT D601K has previously been exploited to demonstrate $K^+$ dependence in the Na,K-ATPase (*Schack et al., 2008*). Collectively, these data suggest that the P-/A-domain site regulation is $K^+$ independent in $P_{IB}$-ATPases, in contrast to classical P-type ATPases. (**e–h**) Evaluation of the effect on selected identified novel inhibitors on activity of protein, as well as survival of mycobacteria and primary human macrophages. (**e**) Effect of two inhibitors (300 μM) on the activity of sCoaT assessed in lipid–detergent solution in the presence of $Zn^{2+}$. For comparison, the commonly used P-type ATPase inhibitor $AlF_4^-$ (500 μM) is included. (**f**) The structure of inhibitors 1 and 2. (**g**) The minimal inhibitory concentration to kill 90% ($MIC_{90}$) of mycobacteria for inhibitors 1 and 2. The mean $MIC_{90}$ value for inhibitor 1 is 18.75 μM, while for inhibitor 2 it is over 50 μM. The values are based on four separate experiments. (**h**) The cytotoxic effect of different concentrations of inhibitors 1 and 2 on primary human macrophages (ATP assay). The standard error of mean (SEM) of nine replicates is shown ($n$ = 9).

## A more potent A-domain modulatory site

A conserved $K^+$ site, which cross-links between the A- and P-domains in E2 states and thereby allosterically stimulates the E2P to E2 process (*Sørensen et al., 2004*; *Schack et al., 2008*), has been suggested to be present also in $P_{IB}$-ATPases (*Sørensen et al., 2004*). However, our new E2 structures and available structures of $P_{IB-1}$- and $P_{IB-2}$-ATPases suggest that the A-/P-domain linker is maintained without $K^+$ in $P_{IB}$-ATPases, and instead is established directly between R273/D601 in sCoaT, as also supported by potassium titration experiments monitoring sCoaT ATPase activity (*Figure 3a–d*). Nevertheless, the A-/P-domain point-of-interaction appears critical for $P_{IB}$-ATPases, as functional characterization of R273A, D601A, and D601K result in a marked reduction of turnover (*Figure 2a*). This differs from similar mutations of classical P-type ATPases, where only minor effects are observed (*Sørensen et al., 2004*; *Schack et al., 2008*). Furthermore, substitution of D601 with glutamate suggests that even the A-/P-domain distance is critical (*Figure 2a*). It is possible that $P_{IB}$-ATPases are more reliant on this particularly tight, ion-independent stabilization, as the A–M1/A-domain linker is absent, and because many other P-type ATPases also have a complementary A-/P-domain interaction (*Figure 1— figure supplement 1c*). Thus, our data indicate that this regulation is a general feature of many P-type ATPase classes, yet featuring unique properties for $P_{IB}$-ATPases.

## New metal-transport blockers

$P_{IB-2}$- and $P_{IB-4}$-ATPases serve as virulence factors and are critical for the disease caused by many microbial pathogens, as underscored by the frequent presence of several redundant genes (*Sassetti and Rubin, 2003*; *Joshi et al., 2006*; *Botella et al., 2011*; *Pi et al., 2016*). In this light and because these P-type ATPases are missing in humans, they represent putative targets for novel antibiotics. The shared mechanistic principles identified here suggest that compounds can be identified that inhibit both $P_{IB}$ groups, for example directed against the common release pathway, thereby increasing efficacy. Indeed, screening of a 20,000-substance library using a complementary in vitro assay, uncovers several compounds that abrogate function of sCoaT and SsZntA (*Figure 3e, f*, data only shown for sCoaT). Furthermore, initial tests of two of these suggest they have a potent effect against mycobacteria, which previously have been shown to be $P_{IB-4}$ dependent for infection (*Patel et al., 2016*); 90% of the mycobacteria were killed at mean concentrations of 18.75 and above 50 μM, respectively, using either of these two separate molecules (*Figure 3g*). In contrast, investigation of cytotoxic effects on primary human macrophages at concentrations up to 25 μM demonstrated considerably less impact on cell survival for both blockers (*Figure 3h*). Evidently downstream in-depth studies, ranging from investigations of the target specificity, the detailed effect on human cells as well as antibiotic potency in human, are required to fully understand the value of these putative $P_{IB-2}$- and $P_{IB-4}$ inhibitors. Nevertheless, the substances outlined here represent promising leads for drug-discovery efforts or to aid the development of tools to manipulate heavy-metal accumulation in plants to prevent accumulation or for enrichment.

## Conclusion

Collectively, the first structure of a $P_{IB-4}$-type ATPase reveals the topology of $P_{IB-4}$-ATPases, displaying an eight helix M-domain configuration, and likely no HMBDs, at least in members without extended N-termini. Major findings include the observation of an ion-release pathway similar as in the related $P_{IB-2}$-ATPases, a previously not observed counterion principle for P-type ATPases, and a unique potassium-independent

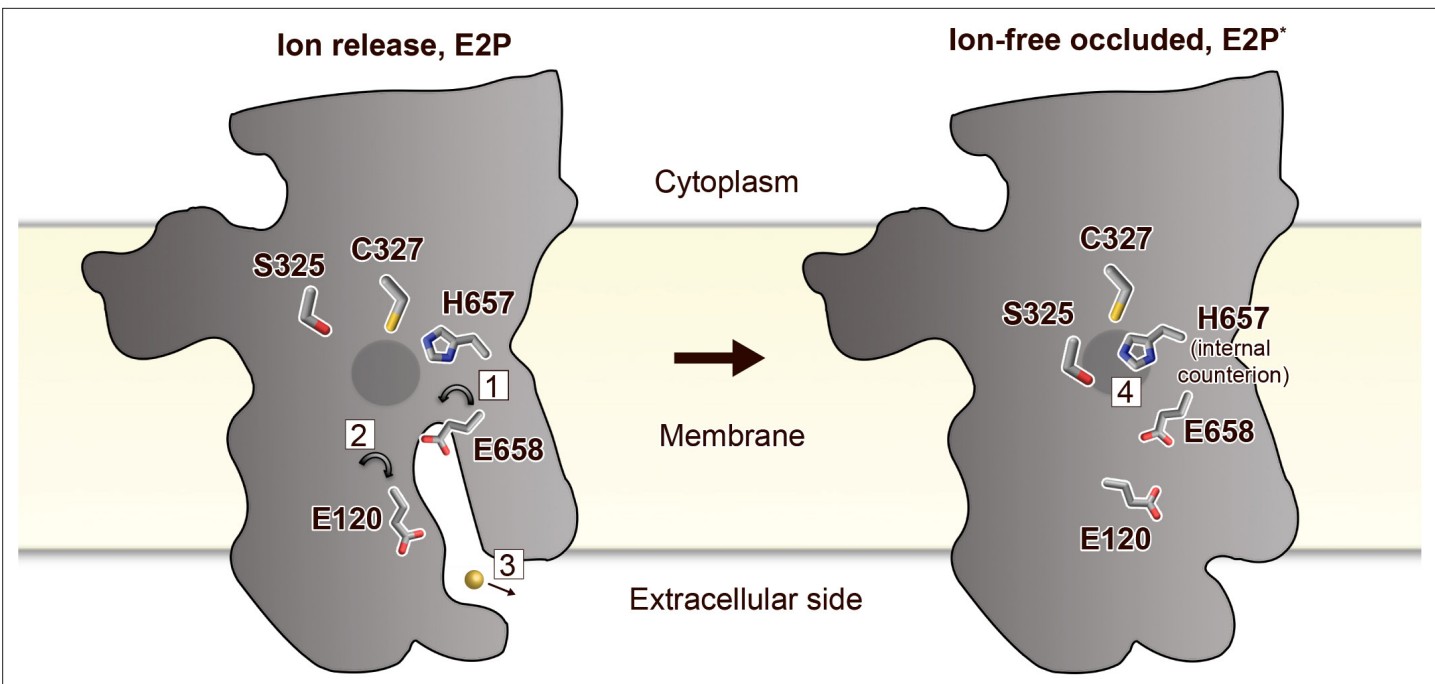

**Figure 4.** Putative ion-release and reocclusion mechanism of $P_{IB-4}$-ATPases. Schematic model illustrating the transmembrane domain (the soluble domains have been removed for clarity) of two separate states, an E2P and an occluded E2P* conformation as the determined structure (E2-BeF$_3^-$), respectively. Zinc or cadmium release from the high-affinity-binding site in the M-domain is likely permitted through re-orientation of E658 (1) in the E1P to E2P transition, thereby lowering the affinity for the occluded ion. E120 serves as a transient linker between the high-affinity-binding site and the outward environment (2). Following ion-release (3) H657 shifts to a sandwiched position between S325 and C327 (4), acting as a built-in counter ion, preventing backtransfer of the released ion, and allowing completion of the reaction cycle.

regulation of the $P_{IB}$-transport cycle (*Figure 4*). Thus, our results significantly increase the understanding of heavy-metal homeostasis in cells. The novel identified putative inhibitors and the partially overlapping mechanistic principles of $P_{IB-2}$- and $P_{IB-4}$-ATPases also open up a novel avenue for development of compounds accessible from outside the cell against these $P_{IB}$ groups, to combat global threats such as multidrug resistance and/or tuberculosis or for biotechnological purposes.

## Materials and methods

**Key resources table**

| Reagent type (species) or resource | Designation | Source or reference | Identifiers | Additional information |
|---|---|---|---|---|
| Gene (*Sulfitobacter* sp. (strain NAS-14.1)) | NAS141_02821 | Synthetic | Uniprot: A3T2G5 | |
| Cell line (*Escherichia coli*) | C41(DE3) | Sigma-Aldrich | | Chemically competent cells |
| Cell line (*Mycobacterium bovis*) | BCG Montreal | | ATCC 35735 | |
| Software, algorithm | Phenix | | RRID:SCR_014224 | https://www.phenix-online.org/ |
| Software, algorithm | ISOLDE | https://doi.org/10.1107/S2059798318002425 | | https://isolde.cimr.cam.ac.uk/ |

*Continued on next page*

*Continued*

| Reagent type (species) or resource | Designation | Source or reference | Identifiers | Additional information |
|---|---|---|---|---|
| Software, algorithm | UCSF ChimeraX | | RRID:SCR_015872 | https://www.cgl.ucsf.edu/chimerax/ |
| Software, algorithm | COOT | | RRID:SCR_014222 | http://www2.mrc-lmb.cam.ac.uk/personal/pemsley/coot/ |
| Software, algorithm | Pymol | | RRID:SCR_000305 | http://www.pymol.org/ |

## Overproduction and purification of sCoAT

Forms of the 72 kDa sCoAT from *Sulfitobacter* sp. NAS14-1 (UniProt ID A3T2G5) were transformed into *E. coli* (C41 strain) cells. The cells were cultured in LB medium at 37°C with shaking at 175 rpm in baffled flasks until the optical density (600 nm) reached 0.6–1, cooled to 18°C, and then induced with 1 mM IPTG for 16 hr. Harvested cells were resuspended in buffer A (1 g cells per 5 ml buffer) containing 20 mM Tris–HCl, pH = 7.6, 200 mM KCl, 20% (vol/vol) glycerol and frozen at −80°C until further use. Cells were disrupted by two runs in a high-pressure homogenizer (Constant System) at 25,000 psi following addition of 5 mM of fresh β-mercaptoethanol (BME), 5 mM $MgCl_2$, 1 mM phenylmethane-sulphonyl fluoride, 2 µg/ml DNase I and Roche protease inhibitor cocktail (1 tablet for 6 l cells). The sample was kept at 4°C throughout the purification. Cellular debris was pelleted via centrifugation at 20,000 × *g* for 20 min. Membranes were isolated by ultracentrifugation for 3 hr at 185,500× *g*, and resuspended in 10 ml buffer B (20 mM Tris–HCl, pH = 7.6, 200 mM KCl, 1 mM $MgCl_2$, 5 mM BME, and 20% [vol/vol] glycerol) per g membranes and frozen at −80°C until further use. The protein concentration in the membranes was estimated using the *Bradford, 1976* assay. Proteins were solubilized through supplementation of 1% (wt/vol) final concentration *n*-dodecyl-β-D-maltopyranoside (DDM) and 3 mg/ml final total protein concentration in buffer B with gentle stirring for 2 hr. Unsolubilized material was removed by ultracentrifugation for 1 hr at 185,500 × *g*. The supernatant was supplemented with imidazole to a final concentration of 30 mM and solid KCl (500 mM final concentration), filtered (0.22 mm), and then applied to 5 ml HiTrap Chelating HP columns (GE Healthcare, protein from 6 l cells per column) charged with $Ni^{2+}$ and equilibrated with four column volumes of buffer C (20 mM Tris–HCl, pH = 7.6, 200 mM KCl, 1 mM $MgCl_2$, 5 mM BME, 150 mg/ml octaethylene glycol mono-dodecyl ether [$C_{12}E_8$] and 20% [vol/vol] glycerol). Proteins were eluted using a gradient, ending with buffer C containing 500 mM imidazole. Eluted protein was assessed using sodium dodecyl sulphate–polyacrylamide gel electrophoresis (SDS–PAGE), and the fractions containing sCoAT concentrated to approximately 20 mg/ml using VivaSpin concentrators (MWCO = 50 kDa). 10 mg concentrated protein was subjected to size-exclusion chromatography using a Superose six gel-filtration column (GE-Healthcare), pre-equilibrated with 50 ml buffer E (20 mM Tris–HCl, pH = 7.6, 80 mM KCl, 1 mM $MgCl_2$, 5 mM BME, 150 mg/ml $C_{12}E_8$ and 20% [vol/vol] glycerol). Fractions containing purified sCoAT were pooled, and concentrated to approximately 10 mg/ml, flash frozen in liquid nitrogen, and stored at −80°C until further use. For the experiments to assess $K^+$ dependence, the buffer E was replaced with 20 mM Tris–HCl, pH = 7.5, 1 mM $MgCl_2$, 5 mM BME, 0.15 mg/ml $C_{12}E_8$, and 20% (vol/vol) glycerol.

## Crystallization

10 mg/ml sCoAT was supplemented with 3 mg/ml (final concentration) DOPC and 6 mg/ml (final concentration) $C_{12}E_8$, incubated at 4°C and stirring for 16–48 hr (modified HiLiDe method *Gourdon et al., 2011a*). Aggregates and insoluble DOPC were then removed by ultracentrifugation at 50,000 ×*g* for 10 min. 2 mM $AlCl_3$ or $BeSO_4$, 10 mM NaF, and 2 mM EGTA (final concentrations) were supplemented and incubated on ice for 30 min. Crystals were grown using the hanging drop vapour diffusion method at 19°C. E2-$AlF_4^-$ crystals were grown with a reservoir solution containing 200 mM $MgCl_2$, 14% (vol/vol) PEG1500, 10 mM tris(2-carboxyethyl)phosphine, 10% (vol/vol) glycerol, 3% 2-methyl-2,4-pentanediol, and 100 mM sodium acetate, pH = 5.0. The E2-$BeF_3^-$ crystals were grown with a reservoir solution containing 200 mM magnesium formate, 14% (vol/vol) PEG5000, 100 mM sodium acetate, pH = 4.0, and 0.5% (vol/vol) 2-propanol was added as an additive. Crystals were

fished using litholoops (Molecular Dimensions), flash cooled in liquid nitrogen, and tested at synchrotron sources. Complete final data sets were collected at the Swiss Light Source, the Paul Scherrer Institute, Villigen, beam line X06SA.

## Structure determination and refinement

Collected data were processed and scaled with XDS (*Table 1*). For the E2-AlF$_4^-$ structure, initial phases were obtained by the MR method using software PHASER (*McCoy et al., 2007*) of the Phenix package (*Liebschner et al., 2019*), and using the AlF$_4^-$-stabilized structure of SsZntA (PDB ID: 4UMW) as a search model. The E2-BeF$_3^-$ structure was solved using the generated E2-AlF$_4^-$ structure as a MR model. Both crystal forms display poor crystal packing between the membrane domains (*Figure 1—figure supplement 2*), deteriorating the quality of the electron density maps in these regions (*Figure 1—figure supplement 3*). In this light, model building of the membrane domains was executed with particular prudence, taking into consideration the connectivity to the well-resolved soluble domains, distinct structural features as well as sequence and structure conservation patterns. Examples of such include the conserved GG motif that forms the kink in MB helix, which is clearly identified also at low length, the SPC motif that twists the M4 helix and the conserved and functionally important well-resolved residue H657 that assisted assigning nearby residues.

Initial manual model building was performed primarily using COOT (*Emsley et al., 2010*). ISOLDE (*Croll, 2018*) in ChimeraX (*Goddard et al., 2018*) was employed for model building and analysis, and was critical for obtainment of the final models with reasonable chemical restraints and low clash score. In particular, ISOLDE's interactive register shifting tool was instrumental in determining the register of the most weakly resolved TM helices. Secondary structure restraints were applied in some flexible regions, also taking into consideration homology to sCoaT and other models.

During final refinements with phenix.refine (*Afonine et al., 2012*), the geometry was restrained in torsion space to ISOLDE's output. Molprobity was exploited for structure validation (*Williams et al., 2018*). The final models are lacking the first 40 residues only, which is shorter than a classical MBD of 67 amino acids. All structural figures were generated using Pymol (*DeLano, 2002*). Statistics for the final models were 96.70, 3.30, 0.20, and 0,74 for E2-BeF$_3^-$ and 93.24, 6.13, 0.63, and 8.31 for E2-AlF$_4^-$ in Ramachandran favored and allowed regions, and for rotamer outliers and clash score, respectively.

Activity assay sCoaT forms were functionally characterized using the Baginski method to assess the amount of released inorganic phosphate (*Baginski et al., 1967*). Briefly, 0.5 μg of purified sCoaT mixed with reaction buffer containing 40 mM MOPS–KOH, pH = 6.8, 5 mM KCl, 5 mM MgCl$_2$, 150 mM NaCl, 0.3 mg/ml C$_{12}$E$_8$, 0.12 mg/ml soybean lipid, 5 mM NaN$_3$, and 0.25 mM Na$_2$MoO$_4$ in a total volume of 50 μl. For metal stimulation assays, different heavy-metal ions or EGTA were supplemented the reaction buffer to a final concentration of 50 μM. For inhibitor screening (see how inhibitors were identified below), different concentrations of inhibitors were added to the reaction buffer containing 50 μM ZnCl$_2$. The samples were then incubated at 37°C with 500 rpm shaking for 5 min, and then supplemented with 5 mM ATP (final concentration) to start the reaction, and incubated at 37°C with 1000 rpm shaking for 10 min. 50 μl freshly prepared stop solution containing 2.5% (wt/vol) ascorbic acid, 0.4 M (vol/vol) HCl, and 1% SDS was then supplemented to stop the reaction and start colour development. 75 μl colour solution (2% [wt/vol] arsenite, 2% [vol/vol] acetic acid, and 3.5% [wt/vol] sodium citrate) was added to the mixture following 10-min incubation at room temperature. Absorbance was measured at 860 nm after another 30-min incubation at room temperature. For the experiments to assess K$^+$ dependence, the reaction buffer was replaced with 40 mM Tris–HCl, pH = 7.5, 5 mM MgCl$_2$, 3.0 mg/ml C$_{12}$E$_8$, and 1.2 mg/ml soybean lipid in a total volume of 50 μl.

## Inhibitor screening

The inhibitor screening experiments were initially carried out on the zinc transporting P$_{IB-2}$-type ATPase ZntA from *Shigella sonnei* (SsZntA). SsZntA was produced and purified as described previously (*Wang et al., 2014*) and the inhibitory effect of approximately 20,000 compounds was assessed by the Chemical biology Consortium Sweden (CBCS). Briefly, the ATPase activity of 0.7 μM highly pure protein was measured in the presence of 50 μM inhibitor through the release of inorganic phosphate (P$_i$) by the Baginski assay (*Baginski et al., 1967*) in a total volume of 200 nl as reported earlier (*Wang et al., 2014*). The inorganic phosphate was detected with Malachite Green reagent (0.005% Carbinol

hydrochloride, 1.7% sulfuric acid, 0.14% ammonium molybdate, 0.025% Triton-X) at an absorbance of 620 nm.

## Minimum inhibitory concentration

*Mycobacterium bovis* bacillus Calmette–Guerin (BCG) Montreal containing the pSMT1-*luxAB* plasmid was prepared as previously described (*Snewin et al., 1999*). Briefly, the mycobacteria were grown in Middlebrook 7H9 broth, supplemented with 10% ADC enrichment (Middlebrook Albumin Dextrose Catalase Supplement, Becton Dickinson, Oxford, UK) and hygromycin (50 mg/l; Roche, Lewes, UK), the culture was washed twice with sterile PBS, and resuspended in broth and then dispensed into vials. Glycerol was added to a final concentration of 25% and the vials were frozen at −80°C. Prior to each experiment, a vial was defrosted, added to 9 ml of 7H9/ADC/hygromycin medium, and incubated with shaking for 72 hr at 37°C. Mycobacteria were then centrifuged for 10 min at 3000 × *g*, washed twice with PBS, and resuspended in 10 ml of PBS. Resazurin microtiter assay was used to determine the minimum inhibitory concentration ($MIC_{90}$) for the inhibitors against the mycobacterial strain. The inhibitors (10 µl) were added to bacterial suspensions (90 µl) on a 96-well plate at a concentration range between 0.4 and 50.0 µM. MIC was determined by the colour change using resazurin (1:10 vol/vol, PrestoBlue Cell viability reagent, Thermo Scientific). MIC was determined after 1 week by adding 10 µl resazurin followed by incubation overnight, corresponding to 90% inhibition.

## Human cytotoxicity assays

Human venous blood mononuclear cells were obtained from healthy volunteers using a Lymphoprep density gradient (Axis-Shield, Oslo, Norway) according to the manufacturer's instructions. To obtain pure monocytes, CD14 microbeads were applied to the cell suspension, washed, and passed through a LS column according to the manufacturer's description (130-050-201, 130-042-401, Miltenyi Biotec, USA). The monocytes were counted (Sysmex), diluted in RPMI 1640 supplemented with 5% FCS, NEAA, 1 mM sodium pyruvate, 0.1 mg/ml gentamicin (11140-035, 111360-039, 15710-49, Gibco, Life Technologies) and 50 ng/ml GM-CSF (215 GM, R&D Systems) and seeded in 96-well plates ($10^5$/well) for a week to differentiate into macrophages. Infection experiments were performed in RPMI 1640 without Gentamicin. The medium was replaced with fresh medium containing 6.3, 12.5, 25, or 50 µM inhibitor or DMSO and incubated 24 hr in 5% $CO_2$ atmosphere. For cytotoxicity measurement, 10 µl 3-(4,5-dimethylthiazol-2-yl)–2,5 diphenyltetrazolium bromide solution (Sigma) was added to each well according to the manufacturer's instructions and analysed in a spectrophotometer at 580 nm. NZX cytotoxicity was further examined by ATPlite assays. Primary macrophages were treated with 6.3, 12.5, 25, or 50 µM inhibitor or DMSO (Sigma) for 24 hr. Cell viability was assessed with cellular ATP levels using ATPlite kit (6016943, Perkin Elmer) compared to untreated controls, according to the manufacturer's instructions.

## MD simulation

The two crystal structures, E2-$AlF_4^-$ and E2-$BeF_3^-$, were inserted into a DOPC (1,2-dioleoyl-sn-glycero-3-phosphocholine) membrane patch using the CHARMM-GUI membrane builder (*Wu et al., 2014*). The membrane positions were predicted by the Orientations of Proteins in Membranes (OPM) server (*Lomize et al., 2012*). During the simulation equilibration phase, position restraints were gradually released from the water and lipids for a total of 30 ns followed by 500 ns non-restrained production runs. Each protein state was simulated in independent repeat simulations starting from a different set of initial velocities, adding up to a sampling total of 500 ns × 4. A Nose–Hoover temperature coupling (*Nosé and Klein, 2006*) was applied using a reference temperature of 310 K. A Parrinello–Rahman pressure coupling (*Parrinello and Rahman, 1981*) was applied with a reference pressure of 1 bar and compressibility of 4.5e−5 $bar^{-1}$ in a semi-isotropic environment. The TIP3P water model was used and the system contained 0.15 M NaCl. The E2-$AlF_4^-$ system was composed of 256 lipids and 29,429 water molecules while E2-$BeF_3^-$ system was composed of 254 lipids and 30,535 water molecules. The systems were equilibrated and simulated using the GROMACS-2021 simulation package (*Abraham et al., 2015*) and CHARMM36 all-atom force fields (*Best et al., 2012*) for the protein and lipids. The membrane domain was used as alignment reference for the root means square deviation and centre-of-mass calculations, and the protein backbone was used as alignment reference for

calculating the root mean square fluctuation. The secondary structure was assessed with the do_dssp tool in GROMACS-2021 (*Abraham et al., 2015*).

Atomic coordinates and structure factors for the sCoaT $AlF_4^-$- and $BeF_3^-$-stabilized crystal structures have been deposited at the Protein Data Bank (PDB) under accession codes 7QBZ and 7QC0. The authors declare no competing financial interests. Correspondence and requests for materials should be addressed to P.G. (pontus@sund.ku.dk).

## Acknowledgements

CG is currently paid by The BRIDGE – Translational Excellence Programme at University of Copenhagen funded by the Novo Nordisk Foundation (NNF18SA0034956). The PhD studies of CG were partly financed by 'The memorial foundation of manufacturer Vilhelm Pedersen and wife – and the Aarhus Wilson consortium'. QH was supported by China Scholarship Council. DRM was funded by Carl Tryggers foundation (CTS 17:22) and MA was funded by a Swedish Research Council Starting Grant (2016-03610). The computations were performed on resources provided by the Swedish National Infrastructure for Computing (SNIC) through the High-Performance Computing Center North (HPC2N) under project SNIC 2018/2-32 and SNIC 2019/2-29. This research was also funded in part by the Wellcome Trust (209407/Z/17/Z) to TC. PG is supported by the following Foundations: Lundbeck, Knut and Alice Wallenberg, Carlsberg, Novo-Nordisk, Brødrene Hartmann, Agnes og Poul Friis, Augustinus, Crafoord as well as The Per-Eric and Ulla Schyberg. Funding is also obtained from The Independent Research Fund Denmark, the Swedish Research Council, and through a Michaelsen scholarship. GM is supported by the Robert A Welch Foundation (AT-1935-20170325 and AT-2073-20210327), the National Institute of General Medical Sciences of the National Institutes of Health (R35GM128704), and the National Science Foundation (CHE-2045984). GG is funded by the Swedish Heart-Lung Foundation (20200378), Alfred Österlunds Foundation, and Royal Physiographic Society of Lund. We are grateful for assistance with crystal screening at PETRA III at DESY, a member of the Helmholtz Association (HGF), beamline P13, and crystal screening and data collection at the Swiss Light Source, the Paul Scherrer Institute, Villigen, beam line X06SA. Access to synchrotron sources was supported by the Danscatt program of the Danish Council of Independent Research. We acknowledge the Chemical Biology Consortium Sweden (CBCS) at Umeå University that performed the ligand screening. For the purpose of Open Access, the authors have applied a CC BY public copyright licence to any Author Accepted Manuscript (AAM) version arising from this submission.

## Additional information

### Funding

| Funder | Grant reference number | Author |
|---|---|---|
| Novo Nordisk Fonden | NNF18SA0034956 | Christina Grønberg |
| The memorial foundation of manufacturer Vilhelm Pedersen and wife - and the Aarhus Wilson consortium | | Christina Grønberg |
| China Scholarship Council | | Qiaoxia Hu |
| Carl Tryggers Stiftelse för Vetenskaplig Forskning | CTS 17:22 | Dhani Ram Mahato |
| Swedish Research Council | 2020-03840 | Magnus Andersson |
| National Institute of General Medical Sciences | R35GM128704) | Gabriele Meloni |
| National Science Foundation | CHE- 2045984 | Gabriele Meloni |
| Swedish Heart-Lung Foundation | 20200378 | Gabriela Godaly |

| Funder | Grant reference number | Author |
|---|---|---|
| Lundbeckfonden | R313-2019-774 | Pontus Gourdon |
| Knut och Alice Wallenbergs Stiftelse | 2020.0194 | Pontus Gourdon |
| Carlsbergfondet | CF15-0542 | Pontus Gourdon |
| Novo Nordisk Fonden | NNF13OC0007471 | Pontus Gourdon |
| Brødrene Hartmann | A29519 | Pontus Gourdon |
| Agnes og Poul Friis Fond | | Pontus Gourdon |
| Augustinus Fonden | 16-1992 | Pontus Gourdon |
| Crafoordska Stiftelsen | 20180652 | Pontus Gourdon |
| Per-Eric and Ulla Schyberg | 38267 | Pontus Gourdon |
| Swedish Research Council | 2016-04474 | Pontus Gourdon |
| The Independent Research Fund Denmark | 9039-00273A | Pontus Gourdon |
| Lundbeckfonden | 218-2016-1548 | Pontus Gourdon |
| Lundbeckfonden | R133-A12689 | Pontus Gourdon |
| Knut och Alice Wallenbergs Stiftelse | 2015.0131 | Pontus Gourdon |
| Carlsbergfondet | 2013_01_0641 | Pontus Gourdon |
| Crafoordska Stiftelsen | 20170818 | Pontus Gourdon |
| Wellcome Trust | 209407/Z/17/Z | Tristan Croll |
| Robert A Welch Foundation | AT-1935-20170325 | Gabriele Meloni |
| National Institute of General Medical Sciences | R35GM128704 | Gabriele Meloni |
| National Science Foundation | CHE-2045984 | Gabriele Meloni |
| Swedish Heart-Lung Foundation | 20200378 | Gabriela Godaly |
| National Supercomputer Centre | 2021/5-362 | Magnus Andersson |
| Robert A Welch Foundation | AT-2073-20210327 | Gabriele Meloni |

The funders had no role in study design, data collection, and interpretation, or the decision to submit the work for publication.

## Author contributions

Christina Grønberg, Conceptualization, Formal analysis, Investigation, Project administration, Supervision, Visualization, Writing - original draft, Writing – review and editing; Qiaoxia Hu, Dhani Ram Mahato, Elena Longhin, Tristan Croll, Formal analysis, Investigation, Visualization, Writing – review and editing; Nina Salustros, Annette Duelli, Pin Lyu, Viktoria Bågenholm, Jonas Eriksson, Komal Umashankar Rao, Domhnall Iain Henderson, Gabriele Meloni, Formal analysis, Investigation, Writing – review and editing; Magnus Andersson, Gabriela Godaly, Formal analysis, Funding acquisition, Investigation, Resources, Supervision, Visualization, Writing – review and editing; Kaituo Wang, Formal analysis, Writing – review and editing; Pontus Gourdon, Conceptualization, Formal analysis, Funding acquisition, Investigation, Project administration, Resources, Supervision, Visualization, Writing - original draft, Writing – review and editing

## Author ORCIDs

Dhani Ram Mahato  http://orcid.org/0000-0002-1121-7761

Pin Lyu http://orcid.org/0000-0003-4532-0430
Gabriele Meloni http://orcid.org/0000-0003-4976-1401
Magnus Andersson http://orcid.org/0000-0002-3364-6647
Gabriela Godaly http://orcid.org/0000-0002-3467-1350
Pontus Gourdon http://orcid.org/0000-0002-8631-3539

## Decision letter and Author response

Decision letter https://doi.org/10.7554/eLife.73124.sa1
Author response https://doi.org/10.7554/eLife.73124.sa2

---

## Additional files

### Supplementary files
• Transparent reporting form

### Data availability
Atomic coordinates and structure factors for the sCoaT AlF4- and BeF3-stabilized crystal structures have been deposited at the Protein Data Bank (PDB) under accession codes 7QBZ and 7QC0.

The following datasets were generated:

| Author(s) | Year | Dataset title | Dataset URL | Database and Identifier |
|---|---|---|---|---|
| Gourdon P, Grønberg C, Hu Q, Mahato DR, Longhin E, Salustros N, Duelli A, Lyu P, Eriksson J, Rao KU, Henderson DI, Meloni G, Andersson M, Croll T, Godaly G, Wang K, Bågenholm V | 2022 | Structure and ion-release mechanism of PIB-4-type ATPases | https://www.rcsb.org/structure/7QBZ | RCSB Protein Data Bank, 7QBZ |
| Gourdon P, Grønberg C, Hu Q, Mahato DR, Longhin E, Salustros N, Duelli A, Lyu P, Bågenholm V, Eriksson J, Rao KU, Henderson DI, Meloni G, Andersson M, Croll T, Godaly G, Wang W | 2022 | Structure and ion-release mechanism of PIB-4-type ATPases | https://www.rcsb.org/structure/7QC0 | RCSB Protein Data Bank, 7QC0 |

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
