## [Editor Report]

This paper presents crystal structures of sCoaT, a heavy metal transporting P-type ATPase. These structures and complementary functional data define the overall fold of this protein and provide insight into several mechanistic features, including a conserved histidine proposed to act as a novel counter-ion during transport. This work will be of interest to biochemists and microbiologists interested in the transport of transition metals, structural biology of membrane proteins and drug development.

---

## [Decision Letter]

**Decision letter after peer review:**

Thank you for submitting your article "Structure and ion-release mechanism of P_IB-4-_type ATPases" for consideration by *eLife*. Your article has been reviewed by 3 peer reviewers, and the evaluation has been overseen by a Reviewing Editor and Olga Boudker as the Senior Editor. The following individuals involved in review of your submission have agreed to reveal their identity: Kazuhiro Abe (Reviewer #1).

The reviewers have discussed their reviews with one another, and the Reviewing Editor has drafted this letter to help you prepare a revised submission. Overall, the reviewers find your manuscript to reflect an interesting and significant study. There are several revisions, however, necessary to strengthen and clarify points, particularly around interpretation of the structures, metal selectivity, and use of MD simulations.

Essential revisions:

Metal selectivity:

1) The characterization of sCoaT activities was carefully done. However, it is unclear why results obtained in this study regarding metal affinity (especially Co2+ and Zn+) is largely different from the previous report (ref 18). Authors just wrote the observed discrepancy is due to the different experimental conditions. Although previous study employed the ATP regenerating coupling assay to determine π release instead of Pi-Mo endpoint assay used in this study, these difference unlikely affect to the determined metal affinity. Are these differences due to the different detergent used (DDM vs C12E8), soybean phospholipids? or different protein sequence? It is helpful for this field to show what is important for measuring ATPase activity of sCoaT. Therefore, please describe possible reason for these discrepancies if authors aware it. Related, XY scattered plot with sigmoidal curve fitting is better than the current version of bar graph format in Suppl Figure 2b. The current form is somewhat misleading for Zn and Co affinity.

2) While discussing the N-terminus, which contains metal-coordinating residues), the authors conclude that “the residues upstream of MA are not essential for catalytic activity” The data shown in Figure 2 support this statement. However, it would be important to determine whether the deletion of the N-terminus affects metal selectivity. If the N-terminus helps to discriminate between Zn and other metals (for example by binding and increasing local concentration of Zn over Co), then the conclusion that “ this level of regulation is absent in PIB-4-ATPases” would not be correct. Thus, it would be helpful and informative to perform the metal titration/activity experiment for the N-terminally truncated construct and compare Zn and Co.

3) Both SsZntA and sCoat transport zinc. As written, it is unclear how structural differences between these transporters affect zinc transport. It would be helpful to state more clearly whether the rates and metal affinity are similar or different for these two pumps.

Interpretation of Structures:

4) To strengthen the presentation of the science, the authors should show electron density associated with the BeF3- and AlF4- ligands and describe how they differentiated the bound ligands from other potential bound substrates (phosphate). The authors should also soften their conclusions regarding the conformational states of the two structures, as they seem to be the same state trapped in the same crystal form.

5) X-ray structures were well modeled as seen in the impressive Rw/Rf values regardless of relatively low-resolution data. However, due to the absence of crystal packing at TM region, TM helices and extracellular side of the protein portion showed poor densities. Although authors overcame this problem of TM modelling by employing ISOLDE, these parts (TM helices and extracellular loops) may be less reliable compared to other well-resolved regions such as cytoplasmic domains. Especially, the extracellular portion of TM6 is important as this portion is directly related to the conclusion of this paper, and hence the electron density map and atomic model should be displayed. In Suppl Figure 4a, TM6 is only shown around H657, and its extracellular side were not shown. In Suppl Figure 4 b, electron density at lower part of TM6 looks sparse, and seems difficult to construct reliable model at this contour level. Related to this issue and also described later, MD trajectories show that the displacement of either E120-E658 or W118-P652 occurs very early stage in all the simulation. This seems to be a consequence of the initial model (crystal structure) being deformed by the MD force field, rather than a conformational change of the enzyme occurring during the transport cycle.

6) It is unclear what does a unique arrangement of the A domain in sCoaT mean (Line212-216), 14 degrees and 3.5A displacement compared to the SsZntA in the same E2.Pi state (Suppl Figure 7). Is it due to the A-domain extension found in sCoaT (L216)? Alternatively, even though TGE motif is superimposable, this 15 degrees tilted A domain conformation in sCoaT is rather similar to E2BeF ground state of SsZntA? If so, comparison with ZntA E2-BeF state is missing and this comparison should be added in the figure (as in Suppl Figure 7), otherwise readers cannot judge whether observed conformation of sCoaT A domain is close to E2P or E2.Pi state of ZntA.

MD Simulations:

7) Structure analysis defined that the BeF-bound form represents a late E2P state (L211, and also concluded in the paragraph starting L280), and AlF-bound form corresponds to E2.Pi transition state (L200). In both states, the relationship between phosphate analog and TGE motif indicates that these are clearly outward-occluded E2.Pi type rather than outward-open E2P ground state. Given that the extracellular gate closure occurred in TM region is coupled to the cytoplasmic domain arrangement, gate opening cannot be expected for both states. Following this logic, the evaluation of gate opening based on the gate-closing crystal structures itself does not make sense. Even though the gate opening is observed in the MD simulation, it is unlikely to occur, at least, does not represent meaningful conformational change in the transport cycle.

Based on this point of logic (for example, one would not expect the outward gate opening in the K-occluded E2-Pi state of NaKATPase), there was a strong suggestion that removing the MD simulation would improve the quality of the manuscript. Doing so would not lessen the conclusions, because you can reach the same conclusion with the simple structural comparison between previously published gate-open SsZntA and present gate-closed sCoaT, to show the metal exit pathway, as these heavy metal pump belongs the same P1B group and possess very similar topology and helix arrangement.

The reviewers discussed whether there could be any usefulness in retaining MD simulations in the manuscript. It was pointed out that you had difficulty modeling the transmembrane domains of the structures, so MD could be an interesting way to look at the stability of the model. It could be that the extracellular gate is not tightly closed in your MODEL, which is what allowed the MD to reveal a plausible ion pathway. This point would be need be strongly clarified in the manuscript

8) Figure 2 and Suppl Figure 8 is confusing. Figure 2d shows E2-BeF (MD) in which E658 does not reach to the E120, but in Figure 2g author showed E2-AlF (MD) result and try to indicate E658-E120 interaction, a 4.5A distance is too far to form sufficiently strong hydrogen bond though. Authors described that gate opening in MD simulation is occurred like showing in Figure 2e, in which lower portion of TM6 is unwound (E2-BeF,MD). However, in Figure 2g and Suppl Figure 8a (E2-AlF, MD) shows different conformational change (probably entire TM6 shift outwardly?).

Other:

9) It is evident that E658 is important for the sCoaT function from the mutagenesis study. However, it is unclear the argument of why E658 is expected to be facing to the metal-binding site in E1 state without having E1 structure.

10) As a correction, E568 in L250 should be E658.

11) The presence of CxxC motif in the MA is an interesting and important finding, as it is generally assumed to be the past of HMBD. To assist the readers and increase clarity, it would be helpful to illustrate the location of this motif in the structure using a cartoon similar to those shown in Suppl Figure 1. Such supplementary cartoon can also show the sequence of the N-terminus. Current illustration of the metal binding residues in the alignment figure using dark green shading obscures rather than highlights these residues.

12) The question about the role of the "platform" is intriguing but simply raising it without offering alternative seems unsatisfactory. Given the flexibility of GlyGly one wonders whether during the transport cycle the "platform" helix straightens and whether the MB'-M1 loop interacts with the N-terminal metal-binding residues to position the N-terminus in the vicinity of the transmembrane domain. Can the platform be a part of the cytosolic gate? The authors may decline to speculate, but would appreciate hearing their thoughts about the possible role of the "platform" helix.

13) In the sentence (lane 314) “uncover several leads that abrogate function…” replace word ‘leads” with “compounds”.

14) It is unclear whether the identified inhibitors specifically abrogate the zinc transport activity of sCoat or equally affect transport of other metals. The authors imply that the broad metal specificity of P1B-4 ATPases is important for virulence. Therefore, it would be of value to perform experiments to show that the inhibitors block the metal-dependent ATPase activity or transport of metals other than zinc. However, such experiments are not essential for this manuscript.

---

## [Author Response]

Essential revisions:Metal selectivity:1) The characterization of sCoaT activities was carefully done. However, it is unclear why results obtained in this study regarding metal affinity (especially Co2+ and Zn+) is largely different from the previous report (ref 18). Authors just wrote the observed discrepancy is due to the different experimental conditions. Although previous study employed the ATP regenerating coupling assay to determine π release instead of Pi-Mo endpoint assay used in this study, these difference unlikely affect to the determined metal affinity. Are these differences due to the different detergent used (DDM vs C12E8), soybean phospholipids? or different protein sequence? It is helpful for this field to show what is important for measuring ATPase activity of sCoaT. Therefore, please describe possible reason for these discrepancies if authors aware it. Related, XY scattered plot with sigmoidal curve fitting is better than the current version of bar graph format in Suppl Figure 2b. The current form is somewhat misleading for Zn and Co affinity.

We thank the reviewers for addressing the matter on metal-specificity. As underscored in the introduction, it is well known that P_IB-4_-ATPases have a broad ion-specificity, ranging from Zn^2+^, Ni^2+^, Cu^+^ and even ca^2+^, and yet these pumps are classically defined as Co^2+^-transporters (for refs see introduction). As such, our findings largely agree with the bulk of the data on P_IB-4_-ATPases, although a previous report was unable to detect Zn^2+^-activity of the here studied target sCoaT. We have expanded the content of Supplementary Figure 2 in an attempt to address this matter. Supplementary Figure 2b is now a hyperbolic plot. In Supplementary Figure 2c, we have explored the possibility of Co^2+^-interference with reducing agent and with the color development in the exploited assay type (known as the Baginski assay). We have also assessed the activity using another assay type known as the Malachite Green Phosphate Assay. None of these parameters significantly affected the results and cannot explain the difference with the previous study. We can only speculate that environmental differences such as lipids, detergents, or other settings may be responsible for the detected variance. Another such aspect that may influence the data is the presence of a his-tag, which naturally binds divalent ions such as Co^2+^.

2) While discussing the N-terminus, which contains metal-coordinating residues), the authors conclude that “the residues upstream of MA are not essential for catalytic activity” The data shown in Figure 2 support this statement. However, it would be important to determine whether the deletion of the N-terminus affects metal selectivity. If the N-terminus helps to discriminate between Zn and other metals (for example by binding and increasing local concentration of Zn over Co), then the conclusion that “ this level of regulation is absent in PIB-4-ATPases” would not be correct. Thus, it would be helpful and informative to perform the metal titration/activity experiment for the N-terminally truncated construct and compare Zn and Co.

We thank the reviewers for this suggestion. We have now conducted preliminary Zn^2+^ and Co^2+^ titration experiments of WT and the deltaN33 form, and conclude that the N-terminus has no major impact on discrimination between these two metal types. The data is available in Supplementary Figure 2d.

3) Both SsZntA and sCoat transport zinc. As written, it is unclear how structural differences between these transporters affect zinc transport. It would be helpful to state more clearly whether the rates and metal affinity are similar or different for these two pumps.

The specific activity and K_M_ for Zn^2+^ is of the same order of magnitude for ZntA and sCoaT. The specific activity in the presence of Zn^2+^ we report for sCoaT is 1.00 ± 0.01 μmol mg^-1^ min^-1^, while the corresponding for SsZntA is 0.59±0.02 μmol mg^-1^ min^-1^ (Wang et al., Nature 2014). In contrast, the affinity for Zn^2+^ and of the SsZntA related pump EcZntA is 10.3±1.9 µM (Mitra and Sharma, Biochemistry 2001), while the equivalent for sCoaT is 4.1 µM, and the low difference (low μM for both pumps), may imply that also in vivo there is little difference in transport capacity.

Interpretation of Structures:4) To strengthen the presentation of the science, the authors should show electron density associated with the BeF3- and AlF4- ligands and describe how they differentiated the bound ligands from other potential bound substrates (phosphate). The authors should also soften their conclusions regarding the conformational states of the two structures, as they seem to be the same state trapped in the same crystal form.

Electron density for the nucleotide binding region is now included in Supplementary Figure 4. The presence of BeF_3_^-^ and AlF_4_^-^, respectively, is supported by strong electron density features in the region, significantly stronger even than certain surrounding main-chain. In agreement with your view, we agree the structures appear highly similar. We have softened the statements in the “Structures in a transition state of dephosphorylation” section.

5) X-ray structures were well modeled as seen in the impressive Rw/Rf values regardless of relatively low-resolution data. However, due to the absence of crystal packing at TM region, TM helices and extracellular side of the protein portion showed poor densities. Although authors overcame this problem of TM modelling by employing ISOLDE, these parts (TM helices and extracellular loops) may be less reliable compared to other well-resolved regions such as cytoplasmic domains. Especially, the extracellular portion of TM6 is important as this portion is directly related to the conclusion of this paper, and hence the electron density map and atomic model should be displayed. In Suppl Figure 4a, TM6 is only shown around H657, and its extracellular side were not shown. In Suppl Figure 4 b, electron density at lower part of TM6 looks sparse, and seems difficult to construct reliable model at this contour level. Related to this issue and also described later, MD trajectories show that the displacement of either E120-E658 or W118-P652 occurs very early stage in all the simulation. This seems to be a consequence of the initial model (crystal structure) being deformed by the MD force field, rather than a conformational change of the enzyme occurring during the transport cycle.

We thank the reviewers for the positive words regarding the careful modelling of the structures! An additional panel for each of the two structures showing the electron density of TM5 and TM6 is now included in Supplementary Figure 4. We hope it will become clear that side-chain features are apparent also for TM6. Regarding the interpretation of the MD simulations, we are as per your request (see also below), now merely exploiting the MD simulations for stability validation of the models. Consequently, the above-mentioned displacement in the MD trajectories is not shown any longer.

6) It is unclear what does a unique arrangement of the A domain in sCoaT mean (Line212-216), 14 degrees and 3.5A displacement compared to the SsZntA in the same E2.Pi state (Suppl Figure 7). Is it due to the A-domain extension found in sCoaT (L216)? Alternatively, even though TGE motif is superimposable, this 15 degrees tilted A domain conformation in sCoaT is rather similar to E2BeF ground state of SsZntA? If so, comparison with ZntA E2-BeF state is missing and this comparison should be added in the figure (as in Suppl Figure 7), otherwise readers cannot judge whether observed conformation of sCoaT A domain is close to E2P or E2.Pi state of ZntA.

We thank the reviewers for pinpointing this matter. As shown in Supplementary Figure 9 the unique arrangement of the A-domain is likely not related to its surface extension, not interfering with the other soluble domains and hence likely not with turn-over. In agreement with this notion, a similar stretch is also present in SERCA. Comparisons also suggest the two sCoaT structures are much more alike the AlF-state of SsZntA rather than the BeF-state of SsZntA (the BeF-state comparison is now included in Supplementary Figure 9). We rather favor one of the two following interpretations. Either the arrangement of the A-domain is unique as previously indicated in the manuscript, or the novel location of the domain is dependent on the crystal packing (this alternative view is now also mentioned in the manuscript).

MD Simulations:7) Structure analysis defined that the BeF-bound form represents a late E2P state (L211, and also concluded in the paragraph starting L280), and AlF-bound form corresponds to E2.Pi transition state (L200). In both states, the relationship between phosphate analog and TGE motif indicates that these are clearly outward-occluded E2.Pi type rather than outward-open E2P ground state. Given that the extracellular gate closure occurred in TM region is coupled to the cytoplasmic domain arrangement, gate opening cannot be expected for both states. Following this logic, the evaluation of gate opening based on the gate-closing crystal structures itself does not make sense. Even though the gate opening is observed in the MD simulation, it is unlikely to occur, at least, does not represent meaningful conformational change in the transport cycle.Based on this point of logic (for example, one would not expect the outward gate opening in the K-occluded E2-Pi state of NaKATPase), there was a strong suggestion that removing the MD simulation would improve the quality of the manuscript. Doing so would not lessen the conclusions, because you can reach the same conclusion with the simple structural comparison between previously published gate-open SsZntA and present gate-closed sCoaT, to show the metal exit pathway, as these heavy metal pump belongs the same P1B group and possess very similar topology and helix arrangement.The reviewers discussed whether there could be any usefulness in retaining MD simulations in the manuscript. It was pointed out that you had difficulty modeling the transmembrane domains of the structures, so MD could be an interesting way to look at the stability of the model. It could be that the extracellular gate is not tightly closed in your MODEL, which is what allowed the MD to reveal a plausible ion pathway. This point would be need be strongly clarified in the manuscript

As per your request, the MD simulations are now merely exploited for stability validation of the models. We have removed the above-mentioned functional interpretation of the *in silico* analyses.

8) Figure 2 and Suppl Figure 8 is confusing. Figure 2d shows E2-BeF (MD) in which E658 does not reach to the E120, but in Figure 2g author showed E2-AlF (MD) result and try to indicate E658-E120 interaction, a 4.5A distance is too far to form sufficiently strong hydrogen bond though. Authors described that gate opening in MD simulation is occurred like showing in Figure 2e, in which lower portion of TM6 is unwound (E2-BeF,MD). However, in Figure 2g and Suppl Figure 8a (E2-AlF, MD) shows different conformational change (probably entire TM6 shift outwardly?).

We thank the reviewers for raising this matter. We hope that the removal of the functional interpretation of the MD analyses will help clarifying how metal transfer from E658 to E120 may take place.

Other:9) It is evident that E658 is important for the sCoaT function from the mutagenesis study. However, it is unclear the argument of why E658 is expected to be facing to the metal-binding site in E1 state without having E1 structure.

As mentioned in the manuscript two separate XAS and mutagenesis studies have suggested that E658 is involved in metal binding (in E1 states). Together with the accumulated understanding on how PIB-ATPases operate, with key roles of the central motifs of TM4, TM5 (in certain subtypes) and TM6, it is reasonable to suggest that this invariant residue (E658) is involved in metal-binding. It is true however that our data does not provide additional evidence for this hypothesis.

10) As a correction, E568 in L250 should be E658.

This sentence has now been removed as it related to observations in the MD simulations.

11) The presence of CxxC motif in the MA is an interesting and important finding, as it is generally assumed to be the past of HMBD. To assist the readers and increase clarity, it would be helpful to illustrate the location of this motif in the structure using a cartoon similar to those shown in Suppl Figure 1. Such supplementary cartoon can also show the sequence of the N-terminus. Current illustration of the metal binding residues in the alignment figure using dark green shading obscures rather than highlights these residues.

We thank the reviewers for this suggestion. An additional panel clarifying the position of the dual cysteines and the arrangement of the N-terminus is now included in the Supplementary Figure 1.

12) The question about the role of the "platform" is intriguing but simply raising it without offering alternative seems unsatisfactory. Given the flexibility of GlyGly one wonders whether during the transport cycle the "platform" helix straightens and whether the MB'-M1 loop interacts with the N-terminal metal-binding residues to position the N-terminus in the vicinity of the transmembrane domain. Can the platform be a part of the cytosolic gate? The authors may decline to speculate, but would appreciate hearing their thoughts about the possible role of the "platform" helix.

We share the view that the role of the platform is elusive. The little available data may well support the outline hypothesis that the platform serves as a cytosolic gate and to provide the appropriate chemical environment for uptake of heavy metals. From a structural biology perspective, the hitherto recovered structures of PIB-ATPases have all demonstrated the platform in a similar configuration, but this analysis is obviously hampered by the fact that only a small fraction of the transport mechanism has been structurally characterized. We anticipate that future studies using complementary techniques will shed further light on this very intriguing matter.

13) In the sentence (lane 314) “uncover several leads that abrogate function…” replace word ‘leads” with “compounds”.

The word ‘leads’ has now been replaced with ‘compounds’ as requested.

14) It is unclear whether the identified inhibitors specifically abrogate the zinc transport activity of sCoat or equally affect transport of other metals. The authors imply that the broad metal specificity of P1B-4 ATPases is important for virulence. Therefore, it would be of value to perform experiments to show that the inhibitors block the metal-dependent ATPase activity or transport of metals other than zinc. However, such experiments are not essential for this manuscript.

We thank the reviewers for this suggestion. As however indicated also by the reviewers, we share the view that this interesting research question is beyond the scope of the current manuscript.